# Epidemiology of African Swine Fever in Piggeries in the Center, South and South-West of Cameroon

**DOI:** 10.3390/vetsci7030123

**Published:** 2020-09-01

**Authors:** Victor Ngu Ngwa, Abdelrazak Abouna, André Pagnah Zoli, Anna-Rita Attili

**Affiliations:** 1School of Veterinary Medicine and Sciences, University of Ngaoundéré, Ngaoundéré P.O. Box 454, Cameroon; abdelrazakabouna1@gmail.com (A.A.); andre.zoli@yahoo.fr (A.P.Z.); 2School of Biosciences and Veterinary Medicine, University of Camerino, Via Circonvallazione 93/95, 62024 Matelica, Italy; annarita.attili@unicam.it

**Keywords:** African swine fever, pigs, prevalence, risk factors, molecular characterization, center-south-southwest Cameroon

## Abstract

African Swine Fever (ASF) is enzootic in Cameroon. A cross-sectional study was conducted in the center, south and south-west regions of Cameroon in order to determine: the knowledge, skills and practices at risk of pig breeders; the prevalence of the disease in piggeries; the genome of the circulating virus. A total of 684 blood samples were collected in 209 farms for RT-PCR and ELISA analyses at the National Veterinary Laboratory (LANAVET) annex in Yaoundé. Prevalences of 15.2% (95CI: 12.5–17.9%) by ELISA, 23.8% (95CI: 20.6–27.0%) by RT-PCR, and 15.2% (95CI: 12.5–17.9%) by ELISA-PCR, were recorded. Of the farmers surveyed, 90% knew about the ASF and 55.3% have already experienced it. The 47.4% of them would not be able to recognize ASF if it occurred and, according to them, the risk of the disease introduction in farms would be 32% linked to the animal health personnel who work on farms. Molecular characterization revealed that only ASF genotype-I variable 19T-RSs is circulating. ASF is still hovering at a risky rate over the pig sector of Cameroon. The control of ASF needs an epidemiological surveillance, a better involvement of all stakeholders, sensitization of breeders and an effective State support for producers.

## 1. Introduction

Faced with the current context of the high rate of urbanization and the strong demographic growth that most African countries are experiencing, the sources of animal protein have become more and more insufficient. Ruminant production is evolving but still cannot meet the needs of the rapidly growing population. To this end, great interest must be paid to the development of short-cycle species, such as pork, in the food self-sufficiency strategies devised by the public authorities. Cameroon has the largest pig population in Central Africa with an estimated herd of around 2,858,548 heads [1]. Unfortunately, for the past two decades, it has faced episodes of African Swine Fever (ASF), a fatal viral disease of swine, against which there is no treatment or vaccine. It is entrenched in Cameroon. Despite the endemicity of the disease in Cameroon and the related annual losses, little epidemiological information about the disease exists. An epidemiological survey was carried out in 2003 by the Directorate of Veterinary Services of MINEPIA (Ministry of Livestock, Fisheries and Animal Husbandry) and reported a high incidence of the order of 12% [2] which has been in clear increase in the years later, recording a prevalence of 15.3% ± 1.6% by ELISA, 22.8% ± 2.2% by nested PCR in 2012, and a prevalence of 20.5% ± 2.4% by real-time PCR in 2014 [3]. In pig farms located in the city of Bafoussam, west region of Cameroon, a prevalence of 23.3% was recorded [4], while an incidence of 0.2% has been reported in the northern zone of the country [2]. From 2010 to 2018, quantitative data, such as the number of outbreaks, deaths and culling of ASF pigs, were collected in the center, south and south-west regions of Cameroon. At National Veterinary Laboratory (LANAVET) level, 602 positive cases were confirmed out of 985 biological samples (blood, sera, and organs) tested during the period 2013 to 2018, given an apparent prevalence of 61.1%. The samples were analyzed by ELISA (386) or by PCR (599), or by ELISA then by PCR (206). The majority (80.8%) of the analyses was requested as part of the investigation of one or more suspected outbreaks of ASF. Next was epidemiological surveillance with 10.1% of occurrences, and then health monitoring of farms came at the last position with 9.1%.

Annual losses in the pig sector are in the range of 26 billion francs CFA (Communauté Financière Africaine) per year [2]. In addition to the direct losses observed, the African Swine Fever Virus generates restrictions on participation in international trade [5]. In Africa, ASFV (African swine fever virus) circulates in sylvatic (transmission between warthogs and soft Argasidae ticks) and domestic (transmission between domestic pigs) cycles, with outbreaks resulting from ASFV spill-over from the sylvatic cycle. This cycle is restricted to eastern, and the southern parts of central Africa. The disease can take several forms depending on the strain of virus and the breed of swine [6]. The causative agent is an arbovirus, a double-stranded, enveloped, icosahedral-symmetric DNA virus that belongs to the family of *Asfarviridae* of *Asfivirus* genus. Complete genome sequencing has identified more than 160 genes in a genomic organization fairly similar to that of poxviruses. Molecular genotyping methods have made it possible to characterize 24 major genotypes that are essential identifiers for understanding the circulation of viral strains and the epidemiology of the corresponding disease [7,8,9,10,11]. Within each genotype, there is a variability of virulence which is still poorly understood. In Cameroon, an analysis of the virus genome from isolates obtained between 1982 and 1987 reveals a small variation between two genomes found [12]. Currently, no vaccine is available and the fight against the disease is based on a rapid diagnosis followed by the implementation of strict sanitary measures.

This study was undertaken to better understand the epidemiology of African Swine Fever in Cameroon. The aims were to: determine the knowledge, skills and practices at risk of pig breeders; estimate the prevalence of the disease by serological and virological investigations in the center, south and south-west regions of Cameroon; carry out a molecular viral characterization for a better understanding of the strain circulating in these regions of Cameroon.

## 2. Materials and Methods

This research does not need an Ethical Approval (with ethical code) because it is based on a normal veterinary epidemiological study, without experimental procedures that are likely to cause pain, suffering, distress or prolonged damage to the consent.

### 2.1. Study Area

The study took place from 2 June to 31 October, 2018 in three regions of Central—South Cameroon, more precisely in the center region (Mfoundi, Nyong and So’o and Lékié divisions); the south region (in the Ocean and Dja and Lobo division), and the south-west region (in Fako division). Figure 1 shows the map of the study area.

The choice of these regions is justified by the fact that they belong to a semi-intensive and intensive production zone with high economic consequences delimited by the ASF control program in Cameroon. The center region is a region with very high risks to ASF, while the south and south-west regions are moderate risk ASF zones. These regions could therefore be considered as a meeting place between animals from various horizons. As for the divisions of these regions, the choice was based on the resurgence of ASF cases in these divisions as well as on the facilities acquired to access and carry out this work such as the availability of MINEPIA staff to provide support in the field, communication channels and the local language spoken.

### 2.2. Type of Study

A cross-sectional epidemiological study was carried out. Prevalence study, acquisition of knowledge and risky practices of breeders in relation to the disease in these regions as well as the molecular characterization of the circulating ASF virus, were investigated. The areas considered in the cross-sectional study are those presenting the highest risk in regard to the occurrence of ASF, according to the history of enzootic events.

### 2.3. Study Methodology

A total of 196 pig farms (minimum) were evaluated using the non-probability quota sampling method [14]. For a workforce of 3 to 5 subjects per farm, the minimum sample size was defined according to the formula:N = (Z^2^ × P × (1 − P))/(e^2^)(1)

Z = significance threshold set at 95%; (e): absolute precision at 5%; P = prevalence estimated at 15.3% [3]; N = 196 farms.

The minimum expected number was 588 animals throughout the study area, but 684 animals were enrolled to increase the precision of the estimated prevalence. The choice of subjects to be sampled was made by the simple random sampling method. Indeed, in animal epidemiology, for communicable diseases, each herd constitutes a group of individuals at shared risk. As a result, the ASF virus should be fairly present in all animals at any given time.

A total of 209 farms were therefore surveyed in 6 divisions: the center region had the largest workforce (135 farms or 65%), followed by the south region (57 farms, 27%) and finally the south-west region (17 farms, 8%). In all, 684 blood samples were collected. The herd size of the farms sampled ranges from 24 to 212 heads.

The numbers of farms and samples collected by region and by division are shown in Table 1.

#### 2.3.1. Samples Collection and Processing

Blood samples were collected by puncturing the jugular vein and centrifuged at 3500 rpm for five minutes using a mobile/fixed centrifuge. Sera were stored at −20 °C until the date scheduled for laboratory analysis. On suspected African swine fever cases, organ harvesting was carried out. After preservation, the carcasses were sent to the LANAVET necropsy room for post mortem examination. The following organs were examined:-Enlarged lymph nodes, with congestive or hemorrhagic “marbling” of the cortical area, or totally hemorrhagic;-The kidneys which, after decapsulation, showed petechiae or even bruising (“turkey egg” appearance);-The spleen with infarction zones and, sometimes, hematomas;-The bladder, skin, larynx, epiglottis and heart, as well as the pleura and peritoneum with petechiae.

These organs were removed, packaged by plastic bags and stored at a temperature of 4 °C. The blood or other liquids present in the carcass during the necropsy were taken in sterile jars as well as pieces of the organs removed.

#### 2.3.2. Laboratory Analyses

The laboratory analyses were twofold: screening for antibodies against the ASF virus in pig serum was carried out using the IDvet ASF indirect ELISA kit (IDvet, Grabels, FRANCE) according to the manufacturer’s instructions; RT-PCR diagnosis was performed on the whole blood, kidney, spleen, lungs, lymph nodes, and liver, following the methodology employed by King et al. [15] partially modified by Wade et al. [16]. Briefly, 2 μL of DNA was added into a 20 μL PCR mix containing 1 × iQ Supermix (Bio-Rad), 400 nM of each primer (King-s 5-CTGCTCATGGTATCAATCTTATCGA-3Gand King-a 5-GATACCACAAGATCRGCCGT-3) and 250 nM of probe (FAM-CCACGGGAGGAATACCAACCCAGTG- TAMRA). The mixture was subjected to 95 °C for 3 min followed by 45 cycles of 95 °C for 10 sec and 58 °C for 30 sec each using the Bio-Rad CFX96 Touch Real-Time PCR Detection System (Bio-Rad, Hercules, CA, USA).

The nucleotide sequences of all the isolates collected in this study were provided by LGC Genomics (Berlin, Germany) to which the purified amplicons (150 in number) were subjected.

To further characterize the ASFV isolates obtained, we amplified, sequenced and analyzed the CVR (central variable region) fragments of the isolates based on tetrameric tandem repeat sequences (TRS) as previously described [15]. Briefly, the C-terminal end of the B646L gene encoding the p72 protein was amplified using P72-U (5′-GGCACAAGTT CGGACATGT-3′) and P72-D (5′-GTACTGTAACGCAG CACAG-3′) primers. Then, a 50 μL reaction, containing 1 × buffer, 2.5 mM MgCl2, 0. Next, 2 mM dNTP, 400 nM of each primer, 0.5 units of Taq polymerase and 4 μL of DNA were used for the amplification. The PCR conditions were as follows: initial denaturation at 95 °C for 5 min, 40 cycles of amplification at 95 °C for 30 sec, 54 °C for 30 sec and 72 °C for 60 sec, and 7 min final extension at 72 °C. The tetrameric tandem repeat sequences within the CVR of the isolates were extracted from the deduced amino acid sequences of the partial B602L gene. Each TRS was transformed into a single letter code utilizing previously published codes for comparison.

#### 2.3.3. Collection of Survey Data and Studied Parameters

In each farm, information was collected using a semi-structured survey form; interviews and direct observations were made. The information collected was related to socio-professional characteristics (sex, age, ethnicity, level of education, main activity, training in breeding), farming techniques (speculation, exploited breed, breeding stage, feeding, health management and therapeutic, etc.), and the level of knowledge about ASF.

### 2.4. Data Analysis

Laboratory results (ELISA), as well as survey data, were input in the Microsoft Excel^®^ 2016 spreadsheet and analyzed using IBM SPSS Statistics v23 software (IBM Corp., Armonk, NY, USA). Prevalences at the divisional and regional level were obtained using the formula below:P = (number of samples tested positive)/(total number of samples) × 100(2)

For the different formulas used, the 95% confidence intervals were determined according to the following formula:CI = [p − 1.96 × √ (P × q / N); p + 1.96 × √ (P × q / N)], with p = prevalence and q = 1 − p.(3)

The phylogenetic trees were built by FASTA and MEGA (Molecular Evolutionary Genetics Analysis) software (Pennsylvania State University, Philadelphia, PA, USA) [17].

## 3. Results

### 3.1. Cross-Sectional Study

Out of the 684 samples collected, 104 were positive by ELISA, 163 by real-time PCR and 104 by both methods. Thus, given an overall prevalence of 15.2% (12.5–17.9; 95% CI) by ELISA; 23.8% (20.6–27.0; 95% CI) by the RT-PCR and 15.2% (12.5–17.9; 95% CI) by ELISA-PCR. The resulting prevalence, for each region is summarized in Table 2.

### 3.2. Knowledge and Risky Practices of Breeders in Relation to ASF

The majority of the farmers (90%) surveyed knew about the ASF, at least by name and 55.3% have already experienced it, among which 35% have had confirmation from LANAVET (Table 3).

In total, 26% of the respondents generally carried out stamping-out and disinfection, while almost 37% of breeders sold “asymptomatic” animals and approximately 21% sold animals showing the first symptoms of the disease.

When asked if they could recognize ASF if it occurred on the farm, 47.4% of farmers replied that they could not recognize ASF; 36.3% of farmers knew the major symptoms of ASF (fever, depression, high morbidity and mortality); 35.8% of breeders confused the symptoms of ASF with those of “diamond skin disease” of pigs (diffuse erythema, conjunctivitis, fever) or other “red pig diseases” (red coloration of the extremities, skin or limbs).

Risky practices of breeders in relation to ASF included all of the activities carried out by farmers, which could constitute a risk for the introduction of the disease on the farm and its spread to surrounding farms and/or localities. This mainly concerns the management of animal husbandry and health aspects on the farm, with a focus on ASF controlling measures. During this study, three types of farm management system were determined, namely breeding in permanent confinement (62%), semi-confinement (35%) and free-ranging (3%).

Table 4 presents the health aspects of pig farms in the study areas. It appears that more than 65% of breeders do not administer any animal care; practically no rodent control or insect control is carried out on the farms.

The footbath is only used by less than 35.4% of breeders, and in most cases it is part of the decor and is ineffective. Quarantine and the infirmary are very rare elements (less than 6.8% on average). Excrement is generally left near farms and used for fertilizing crops (in 95% of cases); 71.7% of breeders consume or dispose of dead animals in the wild; while wastewater generally stagnates in livestock premises until it dries up. Approximately 57.3% of breeders have neighbors who have already experienced ASF. In general, 53.4% of breeders interviewed take no steps to prevent the disease from entering the farm.

### 3.3. Symptoms and Lesions Observed in the Various Cases of ASF

A set of symptoms and lesions were observed in the event of ASF disease by breeders and the authors during the study. The symptoms included conjunctivitis; petechiae at the abdomen, ear flags, and hyperemia of extremities of the limbs (Figure 2); profuse diarrhea; staggering; running nose; decubitus, hyperthermia and general body weakness. Lesions observed include hemorrhagic submaxillary lymphadenitis, splenomegaly, edematous lungs, petechial hemorrhage of the liver, stomach and intestines (Figure 3 and Figure 4).

### 3.4. Molecular Characterization

Viral DNA was detected in all the organs removed. From the 163 samples that tested positive by RT-PCR, it emerged from the sequencing that only one type of strain of virus was present in all the samples. This was the ASF genotype I variable 19 TRSs virus.

## 4. Discussion

This study took place in the center, south and south-west regions of Cameroon that constitutes the major pork production and consumption basins [2], in addition to constituting the geological zones of the plan strategy for preventing and controlling ASF in Cameroon.

An overall prevalence of 15.2% was obtained by the ELISA method and 23.8% by the RT-PCR methodology, in this study. The result is similar to that obtained by Njayou [3], which demonstrated the persistence of the virus and its circulation in the southern part of the country. There is therefore a maintenance of the ASF in the population of domestic pigs in Cameroon though at a lower prevalence compared to other countries of the sub-region, like the Democratic Republic of Congo where a seroprevalence of 37% was recorded in 2020 [18] in the South-Kivu province. In Nigeria a prevalence of 28% was recorded in the south-west region [19], while in other African countries, such as Madagascar and Senegal, a 16% prevalence was obtained at the Alaotra-Mangoro region [20] and a 24% prevalence [21] was recorded, respectively. The prevalence recorded in this study is, however, lower than that obtained in slaughterhouses in countries such as Nigeria [22] and Uganda [23] which implies that this prevalence could have been higher if the study had been conducted at the slaughterhouse level. In addition, it is important to note that in an infected animal, antibodies do not appear until the sixth day after infection [24]. Therefore, pigs at the start of infection might not have been detected by the ELISA method. In all, the variations of prevalence between different regions are attributed to the complexity of the epidemiology of ASF [25,26]. Almost 90% of farmers in the study areas are aware of the ASF. However, a significant percentage does not notify the disease in case of suspicion and for those who notify, less than half of them are aware of the measures imposed by the legislation in force. These facts had previously been reported in Cameroon [3], and constitute one of the weaknesses encountered in the process of controlling ASF in Cameroon.

The lack of organization of pig breeders is believed to be the source of the difficulties in transmitting the information disseminated during awareness-raising tours carried out by the livestock monitoring department on pig diseases. The fact that farmers are coming together does not mean that they are knowledgeable about ASF. The poor functioning of representative structures of breeders (cooperatives, confederations), coupled with the individualism of certain breeders, make the process of helping to revive the sector’s activities ineffective. However, in many countries, including China, it has been proved that recovery assistance, whether financial or logistical support, is an essential element of the disease control and eradication process [27] as it encourages breeders to promptly report cases or suspicions.

Two farming systems were observed with the domination of the confinement type as opposite to the traditional system of practice that predominate in the northern regions of Cameroon [28]. This could be due to the fact that pork is very popular in the southern part of the country, thus, stray pigs could be prone to theft; also, the northern regions have large areas where pigs can wander compared to the southern part where the farms are in close proximity to residential areas, hence being neighbors to each other.

In this study, 44.5% of breeders practice the traditional farming system compared to 67% in the northern part of the country [29]. This factor reduces the risk of a spread of the disease by animal health workers, as breeders who do not receive visits from these workers were less likely to have the disease on their farm, as stipulated by Njayou [3]. Furthermore, 65.5% and 93.2% of the farms evaluated lack a footbath at the entrance and a quarantine facility, respectively, could constitute risks of introduction of the disease. In addition, about 25% of breeders buried or cremated the corpses of animals, while 71.7% consumed or threw them in the wild or on water bodies that also may constitute risks of spread of the disease.

In this study, 90% of the pig farmers visited were able to describe symptoms of suspected cases of ASF dominated by general health and digestive problems. These observations were made by Niang [30] in Senegal (44% for digestive problems). In the Central African Republic, 31.7%, 6.7% and 1.7% of breeders were able to identify digestive, respiratory and nervous symptoms of the disease, while 60% were not able to identify any clinical sign [31]. In Madagascar, on the other hand, 77% of pig breeders could identify at least one clinical sign, and ASF was more incriminated considering skin lesions (51%), nervous symptoms (32%), digestive problems (16%), eye problems (9%) and abortion (3%) [32].

The lesions observed during this study included enlarged and hemorrhagic submaxillary lymph nodes, splenomegaly, hemorrhagic liver with petechial imbibitions, hemorrhagic imbibitions in the kidneys and pulmonary edema. These lesions seemed poorly developed compared to those listed by Remi Adda [33]. The ASFV isolates obtained in 1982, 1985 and 1987 by Ekue [12,34] came from areas located within a 200 km radius in the north-west and west of the country and the genomes of these isolates were indistinguishable by analysis and mapping of restriction enzymes. It was concluded after the study that only one genome was retained, that of the isolate CAM/82, and that the slight variation observed between the latter and the two isolates, CAM/85 and CAM/87, is due to the variation in the number of repetitions present in terminal inverted repetitions. In our study, it emerges that the genomes obtained belong to the same group as that of CAM/82 on the basis of the similar maps of the site of restriction enzymes in their genomes. The isolate CAM/82, which belongs to genotype I, was retained by the scientific community as being specific to Cameroon. However, that is not the case any longer as a variant of this isolate was obtained in the Ivory Coast following an outbreak of ASF in the city of San Pedro [35]. Thus, the virus determined during our study called genotype I variable 19 TT Rs belongs to the same group as that of CAM/82 and is similar to most of the viruses circulating in the sub-region [36].

## 5. Conclusions

Epizootic diseases still limit the development of livestock in Africa. This study revealed that ASF is very much endemic in the study areas in Cameroon. This reflects a flaw in the control measures put in place since the onset of the disease in the areas in 2010, which can be explained by the lack of cooperation from breeders, in order to safeguard their interests among others. Furthermore, this study shows that pig management systems and practices facilitate the spread and persistence of the disease. The ASF virus circulating in the study area have the same genome, which implies that there has not been an introduction of another virus across the borders, thus ruling out the acquisition of virus from the wild. An overall prevalence of 23.8% was determined in the areas of study.

A serious and more effective involvement of all actors in the sector is recommended to fight against ASF and strive for its eradication in these risk regions, in particular, and in Cameroon, in general. In addition, the epidemiological surveillance system must be further boosted, without forgetting the involvement of the scientific community in the process of developing a vaccine against this deadly disease. Ultimately, this study will contribute to the development of appropriate management strategies for the control of ASF in the center, south and south-west regions of Cameroon.

## Figures and Tables

**Figure 1 vetsci-07-00123-f001:**
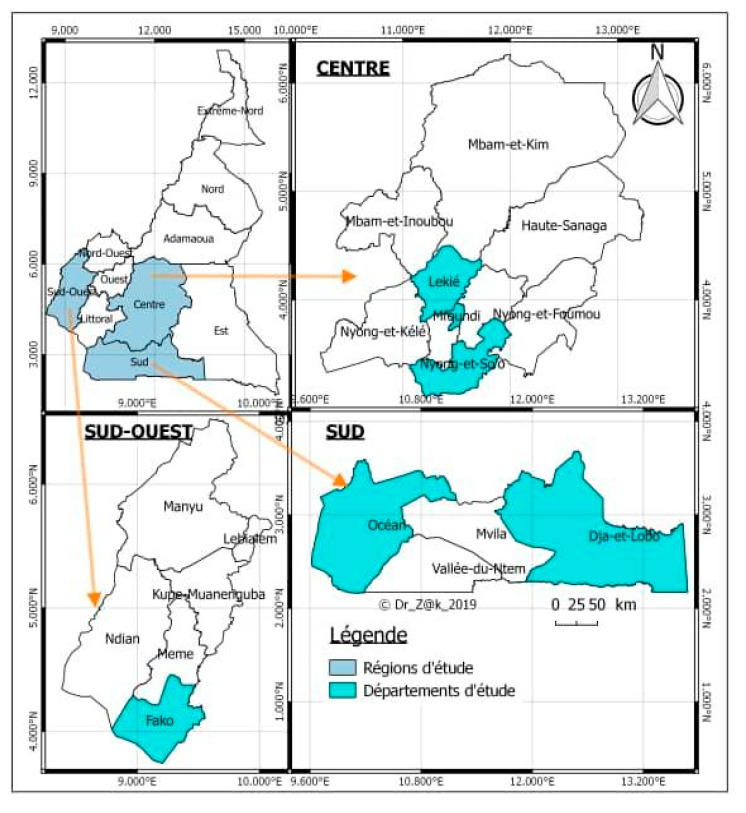
Map showing study areas (Mfoundi, Nyong and So’o and Lékié divisions (departments) in the center (centre) region; Ocean, Dja and Lobo divisions in the south (sud) region, and Fako division in the south-west (sud-ouest) region) of Cameroon. Map of Cameroon was adapted from Wikimedia Commons [13].

**Figure 2 vetsci-07-00123-f002:**
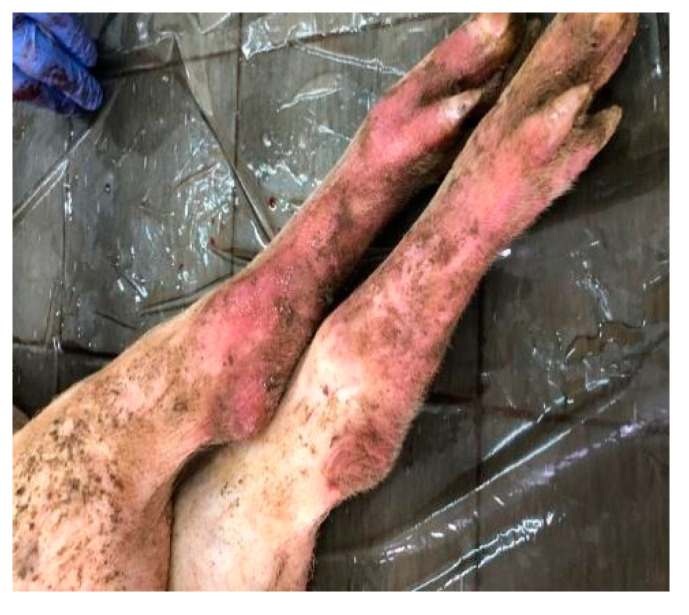
Hyperemia of limbs extremities.

**Figure 3 vetsci-07-00123-f003:**
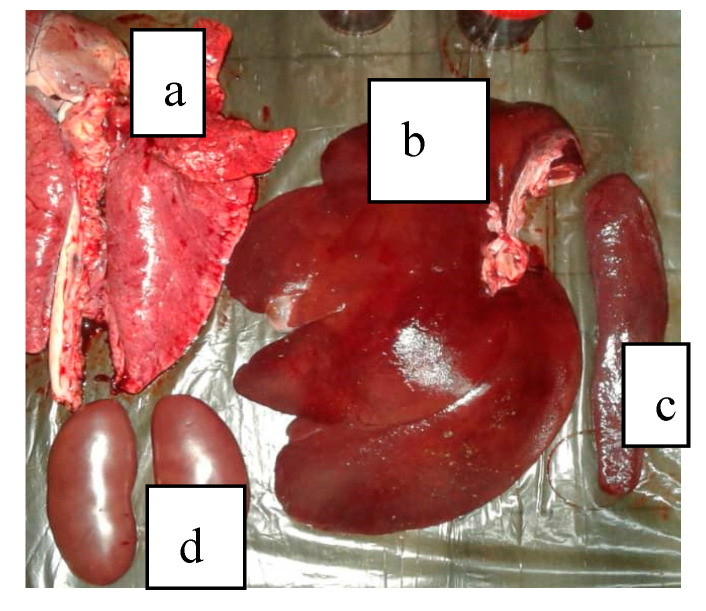
(**a**) Edematous lungs, (**b**) Liver with hemorrhagic imbibitions, (**c**) Enlarged spleen, (**d**) Kidneys with petechiae.

**Figure 4 vetsci-07-00123-f004:**
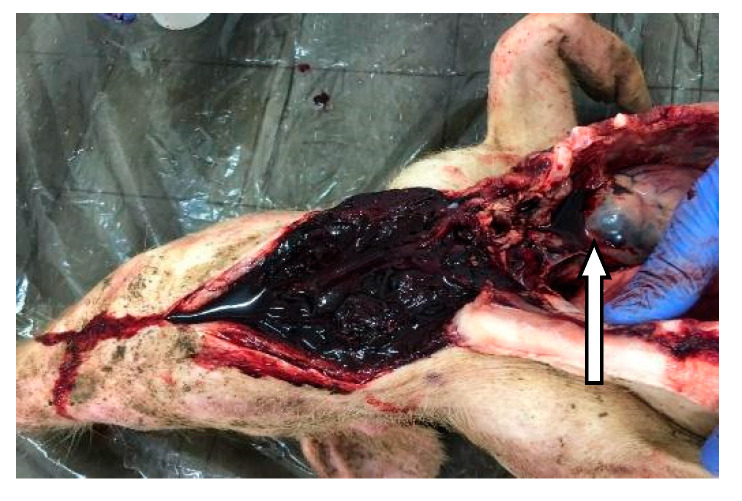
Hypertrophy hemorrhagic submaxillary gland (arrow).

**Table 1 vetsci-07-00123-t001:** Distribution of farms by region and number of samples collected.

Region	Division (Department)	Number of Farms	Number of Samples	Total of Samples Per Region
Center (Centre)	Mfoundi	59	207	441
	Nyong et So’o	31	99	
	Lékié	45	135	
South (Sud)	Océan	34	112	185
	Dja et Lobo	23	73	
South-west (Sud-ouest)	Fako	17	58	58
Total		209	684	684

**Table 2 vetsci-07-00123-t002:** Prevalences (%) by region of ASF in the study areas.

Region	Division	Samples (No)	ELISA ^1^ Prevalence No (%)	PCR ^2^ Prevalence No (%)	ELISA-PCR ^3^ Prevalence No (%)
Center	Mfoundi	207	34 (16.4)	59 (28.5)	34 (16.4)
	Nyong et So’o	99	12 (12.1)	24 (24.2)	12 (12.1)
	Lékié	135	22 (16.3)	40 (29.6)	22 (16.3)
South	Océan	112	16 (14.3)	16 (14.3)	16 (14.3)
	Dja et Lobo	73	11 (15.1)	11 (15.1)	11 (15.1)
South-west	Fako	58	9 (15.5)	13 (22.4)	9 (15.5)
Total		684	104 (15.2)	163 (23.8)	104 (15.2)

^1^ ELISA: *p*-value by division (*p* = 0.950300); by region (*p* = 0.98620); ^2^ PCR: *p*-value by division (*p* = 0.015200); by region (*p* = 0.001400). ELISA-PCR; ^3^
*p*-value by division (*p* = 0.950300); by region (*p* = 0.98620). ASF (African Swine Fever).

**Table 3 vetsci-07-00123-t003:** Summary of information relating to ASF.

Parameters	Modalities	Percentages (%)
Knowledge of ASF	Yes	90
	No	10
Suspected case of ASF in the farm	Yes	55.3
	No	44.6
Laboratory confirmation	Yes	35
	No	65
Post-focus measures	None	9.6
	Stamping out and disinfection	26
	Carcass consumption	14
	Burial and disinfection	9.4
	Sale of “asymptomatic” animals	37
Suspicion or case of ASF among neighbors	Yes	57.3
	No	42.7
Post-case changes/suspicion	None	79.3
	Livestock building	13
	Breeding methods	7.7
Measures taken to avoid entry of the disease into the farm	None	57.7
	Confinement	9
	Ban on visits to the farm	33.2

ASF = African Swine Fever.

**Table 4 vetsci-07-00123-t004:** Health aspects of pig farming.

Parameters	Modalities	Percentages (%)
Animal care	Yes	46.6
	No	43.4
Pest control	Yes	19.2
	No	81.7
Presence of a footbath	Yes	35.4
	No	65.5
Presence of a quarantine	Yes	6.8
	No	93.2
Presence of an infirmary	Yes	6.8
	No	93.2
Excrement management	Close to the farm	7.5
	Close to livestock/Fertilization of crops	77.5
	Collected in a pit/Crop fertilization	13.7
	Thrown far away from the farm	1.3
Management of dead animals	Consumed	71.7
	Buried	23.2
	Thrown away	3.4
	Incinerated	1.5
Wastewater management	Near an entrance to a farm	12.7
	In the farm	71.5
	In a pit	9.2
	Outside the farm	6.6

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
