# Peer review of "Epidemiology of African Swine Fever in Piggeries in the Center, South and South-West of Cameroon"

_vetsci, 2020, doi:10.3390/vetsci7030123_

Round 1

Reviewer 1 Report

African swine fever virus (ASFV) infects wild and domestic pigs causing an acute disease with lethality rates approaching 100%. A better understanding of the disease prevalence and identification of risk factors that impair disease control are of great importance in affected countries. Ngwa et al. present data concerning prevalence of ASF in Cameroon and results obtained from a farmers’ survey. The manuscript merits publication but the following matters should be addressed:

1) Section 2.3.2

The ID Screen® African Swine Fever Indirect ELISA kit detects antibodies against ASFV antigens, not the virus itself. This should be corrected.

The authors should extend the description of the fragments sequenced. Is this a fragment from the B646L gene/p72 protein? Or other fragments were analysed?

2) Section 3.1

The rationale to test samples by ELISA before PCR should be explained (line 155). Since the antibodies appear later in infection, it would make sense to test first for virus presence and after that for antibodies to assess possible infection in the past. It is also not clear from the text how many samples were positive by PCR, ELISA or both. A table presenting this data would be very helpful. Due to the high lethality rates associated with the disease it is very likely that some animals would be positive for virus but negative for antibodies, so samples tested only by ELISA may result in high numbers of false negatives. On the other hand, number of positive samples by ELISA and negative by PCR may give important information concerning the virulence of the strain circulating in the country.

3) Line 244

This isolate nomenclature is confusing. This should be explained in section 2.3.2.

The genotype I refers to p72 sequence. Does the variable refer to the central variable region within the B602L gene and number of tetrameric tandem repeat sequences (TRSs?) This should be explained and discussed further, including comparisons with results published by Wade et al.

Wade A, Achenbach JE, Gallardo C, et al. Genetic characterization of African swine fever virus in Cameroon, 2010-2018. J Microbiol. 2019;57(4):316‐324. doi:10.1007/s12275-019-8457-4

Other points:

Line 51: Most of the ASFV isolates encode for more than 160 genes.

Figure 2: It is not clear which regions are represented in the graph.

Line 276: Do authors mean minority?

Line 278: The authors claim that the antibodies appear at day 6 post infection. This varies greatly from one antigen to another. Does this apply to the IDvet African Swine Fever Indirect ELISA used?

Author Response

Response to Reviewer 1 Comments

Comments and Suggestions for Authors

African swine fever virus (ASFV) infects wild and domestic pigs causing an acute disease with lethality rates approaching 100%. A better understanding of the disease prevalence and identification of risk factors that impair disease control are of great importance in affected countries. Ngwa et al. present data concerning prevalence of ASF in Cameroon and results obtained from a farmers’ survey. The manuscript merits publication but the following matters should be addressed:

1) Section 2.3.2

The ID Screen® African Swine Fever Indirect ELISA kit detects antibodies against ASFV antigens, not the virus itself. This should be corrected.

Response 1: Thank you for your comment. The authors did not specify that indirect ELISA was used to detect antibodies against the ASF virus but at the line 129 the authors followed the reviewer’s suggest and added:  “The laboratory analysis were twofold: screening for antibodies against the ASF virus in pig serum …...”

The authors should extend the description of the fragments sequenced. Is this a fragment from the B646L gene/p72 protein? Or other fragments were analysed?

Response: Thank you for your comment. The authors has accepted to briefly explain the fragments sequenced as described by King et al, 2003 and partially modified by Wade et al, 2019.

Briefly, 2 μl of DNA was added into a 20 μl PCR mix containing 1 × iQ Supermix (Bio-Rad), 400 nM of each primer (King-s 5-CTGCTCATGGTATCAATCTTATCGA-3Gand King-a 5-GATACCACAAGATCRGCCGT-3) and 250 nM of probe (FAM-CCACGGGAGGAATACCAACCCAGTG- TAMRA). The mixture was subjected to 95°C for 3 min followed by 45 cycles of 95°C for 10 sec and 58°C for 30 sec each using the Bio-Rad CFX96 Touch Real-Time PCR Detection System.

2) Section 3.1

The rationale to test samples by ELISA before PCR should be explained (line 155). Since the antibodies appear later in infection, it would make sense to test first for virus presence and after that for antibodies to assess possible infection in the past. It is also not clear from the text how many samples were positive by PCR, ELISA or both. A table presenting this data would be very helpful. Due to the high lethality rates associated with the disease it is very likely that some animals would be positive for virus but negative for antibodies, so samples tested only by ELISA may result in high numbers of false negatives. On the other hand, number of positive samples by ELISA and negative by PCR may give important information concerning the virulence of the strain circulating in the country.

Response 2: Again, thanks for your comment. You are right about testing for the presence of virus first and then antibodies. Nevertheless, due to some reasons beyond our control, both the PCR and ELISA tests were carried out simultaneously.

Secondly, Section 3.2 states the number of samples collected and those positive by ELISA and PCR. However, we failed to include the number of samples that tested positive by both methods. That said, all the samples that were positive by the ELISA method were also positive by PCR technique. Thus, 104 samples tested positive for both the presence of antibodies and the virus. Table 3 has been upgraded accordingly.

3) Line 244

This isolate nomenclature is confusing. This should be explained in section 2.3.2.

Response 3: Thanks for your comment. It was an error. It has been rectified. Corrections are in the text in review mode.

The genotype I refers to p72 sequence. Does the variable refer to the central variable region within the B602L gene and number of tetrameric tandem repeat sequences (TRSs?) This should be explained and discussed further, including comparisons with results published by Wade et al.

Response: Thank you for your comments. Yes, the variable refers to the central variable region within the B602L gene and number of tetrameric tandem repeat sequences.

To a better understanding the characterisation of the ASFV isolates obtained, we amplified, sequenced and analysed the CVR fragments of the isolates based on tetrameric tandem repeat sequences as previously described (Wade et al, 2019). Briefly, the C-terminal end of the B646L gene encoding the p72 protein was amplified using P72-U (5’-GGCACAAGTT CGGACATGT-3’) and P72-D (5’-GTACTGTAACGCAG CACAG-3’) primers.  Then, a 50 μl reaction, containing 1 × buffer, 2.5 mM MgCl2, 0. 2 mM dNTP, 400 nM of each primer, 0.5 units of Taq polymerase and 4 μl of DNA was used for the amplification. The PCR conditions were as follows: initial denaturation at 95°C for 5 min, 40 cycles of amplification at 95°C for 30 sec, 54°C for 30 sec and 72°C for 60 sec, and 7 min final extension at 72°C. The tetrameric tandem repeat sequences within the CVR of the isolates were extracted from the deduced amino acid sequences of the partial B602L gene. Each TRS was transformed into a single letter code utilizing previously published codes for comparison.

In our study, all the ASFV isolates evaluated belonged to the genotype I. Similar results was obtained by Wade et al (2019). However, their study revealed more a diversity among the ASFV isolates and they found three variants of the genotype I virus which were neither regional nor year specific.

The authors added these corrections in the text at the lines 142-152.

Other points:

Line 51: Most of the ASFV isolates encode for more than 160 genes.

Response: Thanks for the information. The sentence has been upgraded.

Figure 2: It is not clear which regions are represented in the graph.

Response: Thanks for your observation. Nevertheless, we have decided to delete Figure 2 for it overlapped with Table 2.

Line 276: Do authors mean minority?

Response: Yes, you said it all. Comments noted and changes effected.

Line 278: The authors claim that the antibodies appear at day 6 post infection. This varies greatly from one antigen to another. Does this apply to the IDvet African Swine Fever Indirect ELISA used?

Response: Thanks again for your comment. We did not claim that antibodies appear at day 6 post infection. It was stated by the author - Sanchez-Vizcaino, J.M. (African Swine Fever. In : Leman A.D.; Straw B.E.; Mengeling W.L.; D’allaire S.; Taylor D.J. Diseases of swine. 7th Edition. London: Wolfe Publishing, 1992, 228-236). Base on this, we assumed that pigs at the start of infection were not detected.

Reviewer 2 Report

The Ngwa et al. performed the epidemiological study of African Swine Fever in pig farms in Cameroon. They conducted the antigen detection using pig serum samples and questionnaire study. These data should be very beneficial to be shared to the readers to understand the ASF situation in Cameroon.

However, this article is just putting the laboratory data and percentage of each category in the questionnaire but did not any statistical analysis. No epidemiological findings relating to the ASF outbreak were obtained at all. The authors must conduct the statistical analysis to seek for potential risk factor(s) of ASF occurrence in the area investigated. I strongly proposed that the authors should calculate the herd prevalence of ASF and check the association between the percentage of positive herd and each variable in the questionnaire.

The authors must select essential figures and tables in the manuscript. Some of the table and figure are overlapped so that the authors must delete either of them (Table 2 vs Figure 2). The contents of Figure 4 is not necessary to be expressed by a figure, but done in text as in Lines 207-209. If there are no intensive discussion points, most of the data relating to symptoms and lesions observed in the various cases of ASF should be taken out. I do not think these are essential to this manuscript.

Regarding the retrospective study, I recommend that the authors should transfer this content to Introduction. Unless performing the comparative analysis between retrospective and cross-sectional study, the former one should not be listed in Results.

Discussion is very long but has poor points to be argued. The authors should select the topics denoted in Discussion with high priority.

There are several technical problems throughout the manuscript; which figure did author use for Absolute precision (Line 100), figure in Table 1 is not correct, and what is “ASF vi labo genotype I variable 19 TT Rs virus” in Line 244.

Author Response

Response to Reviewer 2 Comments

Comments and Suggestions for Authors

The Ngwa et al. performed the epidemiological study of African Swine Fever in pig farms in Cameroon. They conducted the antigen detection using pig serum samples and questionnaire study. These data should be very beneficial to be shared to the readers to understand the ASF situation in Cameroon.

However, this article is just putting the laboratory data and percentage of each category in the questionnaire but did not any statistical analysis. No epidemiological findings relating to the ASF outbreak were obtained at all. The authors must conduct the statistical analysis to seek for potential risk factor(s) of ASF occurrence in the area investigated. I strongly proposed that the authors should calculate the herd prevalence of ASF and check the association between the percentage of positive herd and each variable in the questionnaire.

Response: Thank you for your comment. The authors did not perform a direct ELISA but an indirect ELISA to detect antibodies against the ASF virus but at the line 129 the authors followed the reviewer’s suggest and added:  “The laboratory analysis was twofold: screening for antibodies against the ASF virus in pig serum …...”

For a complete scenario about risk factors, other investigations are in progress by the authors in other divisions of the South West, the littoral, east and northern regions of Cameroon, so data analysis about observed and estimated herd prevalence and Odds ratios (ORs) evaluation as measures of association between outcome and risk factors, with 95% confidence intervals, will be object of further study.

The authors must select essential figures and tables in the manuscript. Some of the table and figure are overlapped so that the authors must delete either of them (Table 2 vs Figure 2). The contents of Figure 4 is not necessary to be expressed by a figure, but done in text as in Lines 207-209. If there are no intensive discussion points, most of the data relating to symptoms and lesions observed in the various cases of ASF should be taken out. I do not think these are essential to this manuscript.

Response: Thanks again for your comments. Yes we agreed with you that some of the tables and figures overlapped. By this, we have deleted figure 2 and figure 4.

Regarding the retrospective study, I recommend that the authors should transfer this content to Introduction. Unless performing the comparative analysis between retrospective and cross-sectional study, the former one should not be listed in Results.

Response: As suggested, from line 45 to 52, the authors moved the propositions about retrospective study to introduction.

Discussion is very long but has poor points to be argued. The authors should select the topics denoted in Discussion with high priority.

Response: Thanks again for your comments. Topics with low priority at the Discussion section have been deleted.

There are several technical problems throughout the manuscript; which figure did author use for Absolute precision (Line 100), figure in Table 1 is not correct, and what is “ASF vi labo genotype I variable 19 TT Rs virus” in Line 244.

Response: Thank you for your comments. We have taking time to amend the technical problems. The authors used 5% for Absolute precision. We believed the figure in Table 1 is correct except proven otherwise. The ASF vi labo genotype I .... is an error. The correct name is ASFV genotype I variable 19 TRS. Again, we are sorry for the technical errors encountered.

Reviewer 3 Report

General

The manuscript contains interesting information about ASF in Cameroon in recent years. It is unfortunately poorly written, mainly due to language difficulties. These may also account for some of the misquotation of references cited, which is of course not acceptable. Much of the writing is difficult to understand. It is also rambling and repetitive, and should be condensed. The important information in this manuscript can be summed up as follows:

  1. As indicated by the authors, ASF is clearly endemic in Cameroon in the area investigated, with outbreaks occurring frequently, characterised by severe disease and high mortality.
  2. The fact that the positive samples were either just positive on PCR for viral genome and negative for antibodies or positive for both virus and antibodies suggests that they all came from pigs during or within a few months after active outbreaks. The pigs that were only positive on PCR were in the early stage of infection before antibodies appear about 7 days post infection, and the other pigs were either in a clinical phase or had possibly recovered, as PCR on appropriate samples (although almost certainly not serum) can be positive for several months, although not necessarily able to be cultured.
  3. The fact that the virus identified in the samples has had a long field presence in Cameroon.
  4. The risk factors that were identified at farm level. They are similar to those identified elsewhere, but interestingly only a small proportion of the farmers allowed their pigs to roam freely. These should be summarised in a table in the results section so that they can be described briefly and in an orderly way, and in the Discussion they could be compared with risk factors described elsewhere, e.g. in Nigeria and other West African countries.

Tables 5 and 6 are not necessary – they could be replaced with a short paragraph indicating that the farmers had reported and the authors had observed clinical signs and pathological lesions suggestive of ASF. The photographs of the lesions can be retained.

An indication of the average herd size of the farms that were sampled would be useful.  

Bearing this outline in mind, the manuscript should be rewritten, with the assistance of a person who is thoroughly competent in English, to present this information clearly and concisely.

Specific

Line 35: I am not sure that ‘rampant’ is the way to describe an endemic situation. ‘Entrenched’ would probably be more appropriate.

Line 44: By ‘slaughtering of ASF’ do the authors mean culling as a result of ASF?

Lines 54-57: This cycle is restricted to eastern, southern and the southern parts of central Africa.

Line 52: ‘arbovirus’ is not a genus of viruses but is a descriptive term derived from ‘Arthropod-Borne viruses’ that is applicable to any virus that has an arthropod biological vector so it should not be italicised or capitalised.

Line 63: Replace ‘pathogenic power’ with ‘virulence’.

Lines 86-87: By health risks do the authors mean risk of ASF?

Line 97: Replace ‘endozootic’ with ‘enzootic’.

Lines 121-122: What is meant by ‘After conditioning’?

I am puzzled as to why RT-PCR was only carried out on sera and not on whole blood. The majority of the ASF virus circulating is associated with the red blood cells, so whole blood in EDTA provides a better sample for detecting virus.

Line 147: Start the sentence: ‘To further characterise the ASFV isolates obtained…’

Line 182: Replace ‘crawl space’ (= the space under the floor of a house) with ‘resting period’

Table 3: I am not sure where the figure of 44.9% of farmers that had already experienced ASF comes from, as Table 3 indicates that only 35% of outbreaks on farms had been confirmed. Authors please to check and correct if necessary, or provide an explanation for the apparent discrepancy.

Also in Table 3, does ‘post-focus changes/suspicion mean changes that were made either after experiencing ASF or suspecting ASF?

Line 202: Do the farmers actually do the stamping-out themselves, or is it done by the government veterinary authority?

Lines 205-209: The figure in Line 19 of the Abstract of 47.4% of farmers who said they would not recognize ASF also does not correlate with what is described in this paragraph, authors please to check and fix or explain.

Lines 207-208: I have never come across a disease of pigs called ‘red mullet’, the authors should please find the correct English name for the disease they are describing. There are various bacterial causes of conjunctivitis in pigs, as well as poor environmental conditions and even mycotoxins.

Lines 225-226: The statement that waste water ‘stagnates in livestock’ cannot be correct – stagnates in livestock premises, perhaps?

Table 5: What is the meaning of the minus sign preceding most of the percentages given?

Table 6 and figures 3 -5: In general there is very little correlation between the table and the figures – the submaxillary lymph node is the only lesion that does correlate with the table.

Lines 266-268: I have no idea where the information about prevalence in the DRC between 2012 and 2015 came from, but the reference cited describes a vaccine trial for ASF in DRC and mentions nothing of the kind, and a paper by the same first author published in 2017 investigated the viruses involved in outbreaks in DRC between 2005 and 2012 and thus also mentions no such thing and the misinformation should be removed. The Bisimwa reference is correctly cited with regard to the overall seroprevalence of 37% but the paper only refers to the South Kivu province of the second largest country in Africa, so this should be clearly stated, as the paper reports considerable variation in prevalence amongst the districts in South Kivu Province, so one might expect much more at national level! The Nigerian and Madagascan studies also refer to very restricted areas; only the reference for Senegal does cover the area from which most ASF outbreaks have been reported. I would suggest that only this reference has any possible relevance for the manuscript under review.

Lines 276-277: While pigs in the early stage of infection would not have antibodies, they would certainly be positive on PCR so should not have been missed in this study.

Lines 317-319: If the pigs are confined, it is unclear how they are able to exercise their coprophagous habits in the fields, and if they really are coprophagous by choice (not true) it would not be necessary to remove the faeces from the pens and put them on the fields for the pigs to eat them. I am also not sure that Montgomery (1921) is an appropriate reference to cite on this topic.

Line 323: The statement that the incubation period for ASF can be up to 40 days is absolute nonsense. The OIE Technical Disease Card for ASF, which I have just downloaded from the OIE website for which the authors give the URL states the following about the incubation period: ‘Incubation period in nature is usually 4-19 days; acute form 3-4 days. For the purposes of the OIE Terrestrial Animal Health Code, the incubation period in Sus scrofa shall be 15 days’. I am sure the authors are confused by the fact that in older versions of the OIE Code and Manual a quarantine period or resting period of 40 days prior to restocking was recommended, although this was an arbitrary figure used for a number of diseases. The policy on the quarantine or resting period prior to restocking is a standard of twice the incubation period for most if not all infectious diseases. However, the 40-day recommendation has been incorporated into the policies of many countries and in Line 182 that is the period given for Cameroon. The maximum incubation period for ASF has was always been considered to be 15 days, and has only recently been extended to 19 as a result of some experimental studies, and it is accepted that the incubation period under natural conditions is usually shorter. The 33 days would therefore normally provide ample time for the pig develop clinical signs. This statement needs to be removed. I am not sure about the basis for the 21 days recommended in the Landrieu reference but as the reference is not specific for ASF it may not be relevant for that disease. The three months of quarantine in Mozambique referred to was in order to import pigs from an ASF-endemic area into South Africa for experimental work and was definitely not science-based but represented an abundance of caution, not to say reluctance, on the part of the SA authorities (it was followed by another three months under strict quarantine in South Africa). The last sentence about detection of latent carriers should be deleted as the reference mentions no such thing in connection with the length of the quarantine period.  

Line 331: What is meant by ‘abatement’?

Lines 337-339: The clinical signs of ASF are generally considered non-specific because they occur in a wide range of febrile diseases and include signs that are not particularly typical of febrile diseases. In acute ASF respiratory signs are most likely to be linked to lung oedema and therefore not typical of respiratory problems such as pneumonia. The fact that the more unusual subacute and chronic manifestations of ASF usually include pneumonia does not appear to be relevant for Cameroon, as the pigs progress rapidly to death.

Lines 351-355: The lesions described and for which photographs were provided are typical of acute severe ASF. I do not have access to reference 35 but reference 36 makes a very general reference to ‘functional and congestive-haemorrhagic disorders of the digestive and respiratory systems’ caused by acute ASF (in fact such disorders can and usually do affect all the systems) but no mention is made of the pathological lesions in the organs that the authors mention so it should definitely be omitted. Misquoting literature is unacceptable.

Lines 356-373: The paragraph is rather confusing and should be shortened to state clearly the finding that the virus found in this study has had a field presence in Cameroon since 1982. It may be of interest to you that the CAM 1982 variant was isolated from an outbreak of ASF in the port of San Pedro in Côte d’Ivoire in 2014. I have attached the paper for your interest.

Author Response

Response to Reviewer 3 Comments

General

The manuscript contains interesting information about ASF in Cameroon in recent years. It is unfortunately poorly written, mainly due to language difficulties. These may also account for some of the misquotation of references cited, which is of course not acceptable. Much of the writing is difficult to understand. It is also rambling and repetitive, and should be condensed. The important information in this manuscript can be summed up as follows:

  1. As indicated by the authors, ASF is clearly endemic in Cameroon in the area investigated, with outbreaks occurring frequently, characterised by severe disease and high mortality.
  2. The fact that the positive samples were either just positive on PCR for viral genome and negative for antibodies or positive for both virus and antibodies suggests that they all came from pigs during or within a few months after active outbreaks. The pigs that were only positive on PCR were in the early stage of infection before antibodies appear about 7 days post infection, and the other pigs were either in a clinical phase or had possibly recovered, as PCR on appropriate samples (although almost certainly not serum) can be positive for several months, although not necessarily able to be cultured.
  3. The fact that the virus identified in the samples has had a long field presence in Cameroon.
  4. The risk factors that were identified at farm level. They are similar to those identified elsewhere, but interestingly only a small proportion of the farmers allowed their pigs to roam freely. These should be summarised in a table in the results section so that they can be described briefly and in an orderly way, and in the Discussion they could be compared with risk factors described elsewhere, e.g. in Nigeria and other West African countries.

Tables 5 and 6 are not necessary – they could be replaced with a short paragraph indicating that the farmers had reported and the authors had observed clinical signs and pathological lesions suggestive of ASF. The photographs of the lesions can be retained.

Response 1: Thanks for your observation, Tables 5 and 6 has been deleted and replaced with a short paragraph.

An indication of the average herd size of the farms that were sampled would be useful.  

Response 2: Thanks for your comment. The herd size of the farms sampled ranges from 24 to 212 heads.

Bearing this outline in mind, the manuscript should be rewritten, with the assistance of a person who is thoroughly competent in English, to present this information clearly and concisely.

Response 3: Thank you for your comment. We have revised the manuscript accordingly.

Specific

Line 35: I am not sure that ‘rampant’ is the way to describe an endemic situation. ‘Entrenched’ would probably be more appropriate.

Response 4: Thanks for your observation. Comment noted and change effected.

Line 44: By ‘slaughtering of ASF’ do the authors mean culling as a result of ASF?

Response 5: Yes, you said it all. Comments noted and changes effected.

Lines 54-57: This cycle is restricted to eastern, southern and the southern parts of central Africa.

Response 6: Thanks for the comment. Statement included in the text accordingly.

Line 52: ‘arbovirus’ is not a genus of viruses but is a descriptive term derived from ‘Arthropod-Borne viruses’ that is applicable to any virus that has an arthropod biological vector so it should not be italicised or capitalised.

Response 7: Again, thanks for your observation. Comment noted and change effected.

Line 63: Replace ‘pathogenic power’ with ‘virulence’.

Response 8: Thank you again for the observation. Comment noted and change effected.

Lines 86-87: By health risks do the authors mean risk of ASF?

Response 9: Yes, the authors meant risk of ASF. The sentence has been revised accordingly

Line 97: Replace ‘endozootic’ with ‘enzootic’.

Response 10: Done! Thank you for the observation.

Lines 121-122: What is meant by ‘After conditioning’?

Response 11: Thanks for your observation. We meant after preservation.

I am puzzled as to why RT-PCR was only carried out on sera and not on whole blood. The majority of the ASF virus circulating is associated with the red blood cells, so whole blood in EDTA provides a better sample for detecting virus.

Response 12: Thank again for the observation. Sorry for the error. RT-PCR was carried out on whole blood.

Line 147: Start the sentence: ‘To further characterise the ASFV isolates obtained…’

Response 13: Done! Thanks for the observation.

Line 182: Replace ‘crawl space’ (= the space under the floor of a house) with ‘resting period’.

Response 14: The authors followed the suggestion.

Table 3: I am not sure where the figure of 44.9% of farmers that had already experienced ASF comes from, as Table 3 indicates that only 35% of outbreaks on farms had been confirmed. Authors please to check and correct if necessary, or provide an explanation for the apparent discrepancy.

Response 15: Thanks for your comments! It is 55.3% of farmers had already experienced ASF and not 44.9%, and 35% of the suspected cases were confirmed in the laboratory.

Also in Table 3, does ‘post-focus changes/suspicion mean changes that were made either after experiencing ASF or suspecting ASF?

Response 16: Yes, you are right!

Line 202: Do the farmers actually do the stamping-out themselves, or is it done by the government veterinary authority?

Response 17: Yes, farmers can do the stamping-out themselves but it’s usually done by the government veterinary authority.

Lines 205-209: The figure in Line 19 of the Abstract of 47.4% of farmers who said they would not recognize ASF also does not correlate with what is described in this paragraph, authors please to check and fix or explain.

Response 18: Thanks again for the observation. It is a mistake. The correct percentage is 47.4%.

Lines 207-208: I have never come across a disease of pigs called ‘red mullet’, the authors should please find the correct English name for the disease they are describing. There are various bacterial causes of conjunctivitis in pigs, as well as poor environmental conditions and even mycotoxins.

Response 19: Thanks for your observation. Again, it is an error. The disease in question here is swine erysipelas or diamond skin disease of pigs. Correction has been effected.

Lines 225-226: The statement that waste water ‘stagnates in livestock’ cannot be correct – stagnates in livestock premises, perhaps?

Response 20: Yes, you said it all. Thanks for the observation.

Table 5: What is the meaning of the minus sign preceding most of the percentages given?

Response 21: Table 5 has been deleted, as suggested.

Table 6 and figures 3 -5: In general there is very little correlation between the table and the figures – the submaxillary lymph node is the only lesion that does correlate with the table.

Response 22: Also, table 6 has been deleted, as suggested.

Lines 266-268: I have no idea where the information about prevalence in the DRC between 2012 and 2015 came from, but the reference cited describes a vaccine trial for ASF in DRC and mentions nothing of the kind, and a paper by the same first author published in 2017 investigated the viruses involved in outbreaks in DRC between 2005 and 2012 and thus also mentions no such thing and the misinformation should be removed. The Bisimwa reference is correctly cited with regard to the overall seroprevalence of 37% but the paper only refers to the South Kivu province of the second largest country in Africa, so this should be clearly stated, as the paper reports considerable variation in prevalence amongst the districts in South Kivu Province, so one might expect much more at national level! The Nigerian and Madagascan studies also refer to very restricted areas; only the reference for Senegal does cover the area from which most ASF outbreaks have been reported. I would suggest that only this reference has any possible relevance for the manuscript under review.

Response 23: Thanks again for your comments. We accept your suggestions for Senegal. But the authors believed also that the regional studies carryout in Nigeria, Madagascar and South-kivu province could be relevant references.

Lines 276-277: While pigs in the early stage of infection would not have antibodies, they would certainly be positive on PCR so should not have been missed in this study.

Response 24: Again, you are right. That’s why we have more positive cases of ASF by PCR as compared to the ELISA method.

Lines 317-319: If the pigs are confined, it is unclear how they are able to exercise their coprophagous habits in the fields, and if they really are coprophagous by choice (not true) it would not be necessary to remove the faeces from the pens and put them on the fields for the pigs to eat them. I am also not sure that Montgomery (1921) is an appropriate reference to cite on this topic.

Response 25: Thanks for your comments. We have deleted the paragraph.

Line 323: The statement that the incubation period for ASF can be up to 40 days is absolute nonsense. The OIE Technical Disease Card for ASF, which I have just downloaded from the OIE website for which the authors give the URL states the following about the incubation period: ‘Incubation period in nature is usually 4-19 days; acute form 3-4 days. For the purposes of the OIE Terrestrial Animal Health Code, the incubation period in Sus scrofa shall be 15 days’. I am sure the authors are confused by the fact that in older versions of the OIE Code and Manual a quarantine period or resting period of 40 days prior to restocking was recommended, although this was an arbitrary figure used for a number of diseases. The policy on the quarantine or resting period prior to restocking is a standard of twice the incubation period for most if not all infectious diseases. However, the 40-day recommendation has been incorporated into the policies of many countries and in Line 182 that is the period given for Cameroon. The maximum incubation period for ASF has was always been considered to be 15 days, and has only recently been extended to 19 as a result of some experimental studies, and it is accepted that the incubation period under natural conditions is usually shorter. The 33 days would therefore normally provide ample time for the pig develop clinical signs. This statement needs to be removed. I am not sure about the basis for the 21 days recommended in the Landrieu reference but as the reference is not specific for ASF it may not be relevant for that disease. The three months of quarantine in Mozambique referred to was in order to import pigs from an ASF-endemic area into South Africa for experimental work and was definitely not science-based but represented an abundance of caution, not to say reluctance, on the part of the SA authorities (it was followed by another three months under strict quarantine in South Africa). The last sentence about detection of latent carriers should be deleted as the reference mentions no such thing in connection with the length of the quarantine period.  

Response 26: Thanks again for your lengthy observations. We agreed with your submission. Statements and paragraph have been deleted.

Line 331: What is meant by ‘abatement’?

Response 27: Thanks for the observation. ‘Abatement’ means dejection. Nevertheless, statement has been deleted.

Lines 337-339: The clinical signs of ASF are generally considered non-specific because they occur in a wide range of febrile diseases and include signs that are not particularly typical of febrile diseases. In acute ASF respiratory signs are most likely to be linked to lung oedema and therefore not typical of respiratory problems such as pneumonia. The fact that the more unusual subacute and chronic manifestations of ASF usually include pneumonia does not appear to be relevant for Cameroon, as the pigs progress rapidly to death.

Response 28: Ok! Thanks for the information. The Authors agree.

Lines 351-355: The lesions described and for which photographs were provided are typical of acute severe ASF. I do not have access to reference 35 but reference 36 makes a very general reference to ‘functional and congestive-haemorrhagic disorders of the digestive and respiratory systems’ caused by acute ASF (in fact such disorders can and usually do affect all the systems) but no mention is made of the pathological lesions in the organs that the authors mention so it should definitely be omitted. Misquoting literature is unacceptable.

Response 29: Thanks for your observation. Reference has been deleted as suggested.

Lines 356-373: The paragraph is rather confusing and should be shortened to state clearly the finding that the virus found in this study has had a field presence in Cameroon since 1982. It may be of interest to you that the CAM 1982 variant was isolated from an outbreak of ASF in the port of San Pedro in Côte d’Ivoire in 2014. I have attached the paper for your interest.

Response 30: Thanks again for your comments and for the reference. We have shortened the paragraph and state clearly the findings.

Round 2

Reviewer 1 Report

The authors answered most of my queries, however the manuscript would benefit from further shortening, particularly the Discussion section.

Author Response

Response to Reviewer 1 Comments

Thank you for your comment. The authors has accepted and have shorten the manuscript accordingly, especially the Discussion section.

Reviewer 2 Report

Though the authors tried to respond to inquired from myself and the other reviewer, they would not like to have an idea following my critical inquiry. The authors pointed out the proportions of factors in the variables. However, no significant factors for ASFV infection will be identified under the 15-25 % positive rates unless calculating the ORs among the infected and non-infected population. If the statistical analysis will be conducted by the other group, this manuscript is just describing the prevalence but is not describing the epidemiological findings. The authors should not separate the results in two publications. If the authors must separate the results and put only the results of the prevalence, all the tables and figures expect Table 1 and Table 2 are unnecessary for publication. According to the small amount of the results, the manuscript with only Table 1 and Table 2 could be published as Short Communication.

Author Response

Response to Reviewer 2 Comments (Second round)

Though the authors tried to respond to inquired from myself and the other reviewer, they would not like to have an idea following my critical inquiry. The authors pointed out the proportions of factors in the variables. However, no significant factors for ASFV infection will be identified under the 15-25 % positive rates unless calculating the ORs among the infected and non-infected population. If the statistical analysis will be conducted by the other group, this manuscript is just describing the prevalence but is not describing the epidemiological findings. The authors should not separate the results in two publications. If the authors must separate the results and put only the results of the prevalence, all the tables and figures expect Table 1 and Table 2 are unnecessary for publication. According to the small amount of the results, the manuscript with only Table 1 and Table 2 could be published as Short Communication.

Response: Thank you for the comments formulated in order to improve the article. The manuscript was a cross-sectional study carried out in the Center, South and South-West regions of Cameroon in order to determine the prevalence of African Swine Fever, and the genome of the circulating virus in these geographical areas of African continent. The Authors think that the descriptive epidemiological data and molecular epidemiology could contribute to fulfil the knowledge on this topic and to be a primary research tool for systematic review and meta-analysis studies.

Reviewer 3 Report

The authors have made all the necessary changes, thank you for the careful attention given to the task. The English is greatly improved. There are still some errors (the main examples below should receive attention) but I assume the article will receive further language editing before publication. There is just one non-language error in the Abstract that still needs to be corrected. 

Lines 17-19 in the Abstract: In the response from the authors they state that the figure of 44.9% is erroneous and should be 55.3% - this has been corrected in Line 188 and must be corrected in the Abstract as well.

Line 67: ‘Nevertheless’ should be deleted, it has no meaning in this context. The sentence could be rewritten as follows: This study was undertaken to better understand the epidemiology of African swine fever in Cameroon.

Line 136: Laboratory analyses (the plural of analysis, since two were carried out).

Line 206: Replace ‘wandering’ with ‘free-ranging’ (it is the correct English word to describe ‘divagation’ in pigs).

Author Response

The authors have made all the necessary changes, thank you for the careful attention given to the task. The English is greatly improved. There are still some errors (the main examples below should receive attention) but I assume the article will receive further language editing before publication. There is just one non-language error in the Abstract that still needs to be corrected. 

Lines 17-19 in the Abstract: In the response from the authors they state that the figure of 44.9% is erroneous and should be 55.3% - this has been corrected in Line 188 and must be corrected in the Abstract as well.

Response 1: Thank you for the observation. Error has been corrected.

Line 67: ‘Nevertheless’ should be deleted, it has no meaning in this context. The sentence could be rewritten as follows: This study was undertaken to better understand the epidemiology of African swine fever in Cameroon.

Response 2: Thanks again, the sentence has been rephrase according to your suggestion.

Line 136: Laboratory analyses (the plural of analysis, since two were carried out).

Response 3: Error corrected. Thanks for the observation.

Line 206: Replace ‘wandering’ with ‘free-ranging’ (it is the correct English word to describe ‘divagation’ in pigs).

Response 4: Thanks again for your observation. We accept your suggestion and have effect the change.
